# Learning About Objects
# by Learning to Interact with Them

**Martin Lohmann[1], Jordi Salvador[1], Aniruddha Kembhavi[1,2], Roozbeh Mottaghi[1,2]**

[1]PRIOR @ Allen Institute for AI    [2]University of Washington
{martinl,jordis,anik,roozbehm}@allenai.org
https://prior.allenai.org/projects/learning_from_interaction

## Abstract

Much of the remarkable progress in computer vision has been focused around fully supervised learning mechanisms relying on highly curated datasets for a variety of tasks. In contrast, humans often learn about their world with little to no external supervision. Taking inspiration from infants learning from their environment through play and interaction, we present a computational framework to discover objects and learn their physical properties along this paradigm of *learning from interaction*. Our agent, when placed within the near photo-realistic and physics-enabled AI2THOR environment, interacts with its world and learns about objects, their geometric extents and relative masses, without any external guidance. Our experiments reveal that this agent learns efficiently and effectively; not just for objects it has interacted with before, but also for novel instances from seen categories as well as novel object categories.

## 1   Introduction

Over the past few years, the computer vision community has witnessed remarkable breakthroughs on a variety of tasks such as image classification [31], object detection [49] and semantic segmentation [9]. Much of this progress can be attributed to the re-emergence of deep learning in an era of tremendous compute, and importantly, the availability of large annotated datasets that enable models to be trained in a fully supervised learning paradigm. While recent works [25, 10] have shown an impressive ability to train visual representation stacks in a self-supervised manner, downstream applications using these representations continue to be trained with fully supervised methods [23].

In stark contrast, humans often learn about their world with little or even no external supervision. For instance, infants learn about objects in their physical environment and their behaviors just by observing [4] and interacting [21] with them. Inspired by these studies, we propose a computational approach to discover objects and learn their physical properties in a self-supervised setting along this paradigm of *Learning from Interaction*.

Learning about objects by interacting entails the following steps: First, given an environment, the learning agent must pick a location in space, perhaps an object, to interact with. Second, the agent must determine the nature of this interaction (for instance, pushing, lifting, throwing, etc). Third, the result of this interaction must be interpreted solely via visual feedback and with no external supervision. And finally, it must iterate over these steps effectively and efficiently with the goal of learning about objects and their attributes. For instance, an agent attempting to learn to estimate the mass of objects based on their appearance may find it beneficial to interact with a diverse set of objects, pick or push them and finally acquire feedback via a combination of the force applied and the resultant displacement on the objects.

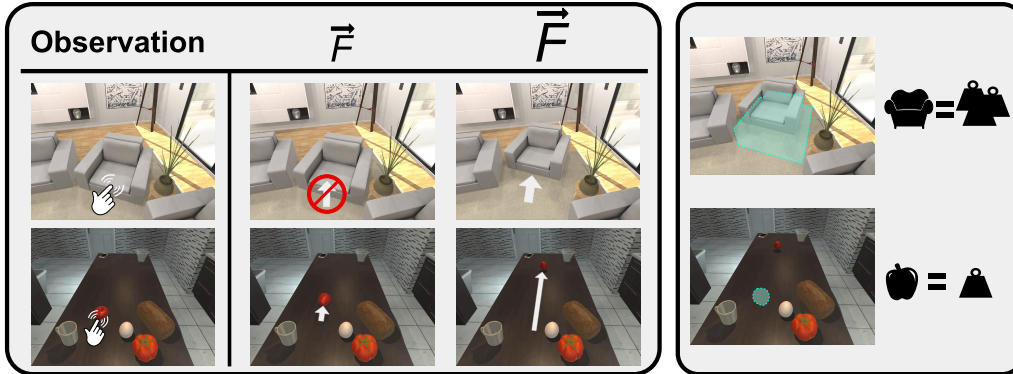

Figure 1: Our goal is to learn geometric extents (segmentation) and masses of objects in a self-supervised fashion by interacting with the surrounding world. The agent should learn not only which parts of the current observation are interactable but also how to interact with them. For example, a small force may not move a sofa but moves an apple, which enables estimating the object properties.

In this work, we present an agent that learns to locate objects, predict their geometric extents as well as relative masses merely by interacting with its environment, with no external supervisory signal (Figure 1). Our proposed model learns to predict what it should interact with and what forces should be applied to those points. The interaction results in a sequence of raw observations, which are used as supervision for training a CNN model. The raw observations enable computing self-supervised losses for the interaction point, the amount of force, and for clustering points that move coherently upon interaction and so probably belong to a single object. To stabilize training and make it more efficient, we use a memory bank with prioritized sampling which is inspired by prioritized replay and self-paced learning.

We train and evaluate our agent within the AI2THOR [30] environment, a near photo-realistic virtual indoor environment of 120 rooms such as kitchens and living rooms with more than 2,000 object instances across 125 categories. Importantly, AI2THOR is based on a physics engine which enables objects to have physical properties such as mass, friction and elasticity and for objects and scenes to interact realistically with each other. Our agent is placed in this world with no prior knowledge (apart from the inductive bias of a CNN and a self-supervision module), interacts with this world by applying forces, starts learning about the presence of objects via their displacement and eventually learns to visually estimate their attributes. Experimental evaluations show that our model obtains promising results for novel instances of seen object categories as well as unseen object categories.

## 2 Related Work

We now present a series of works that explore the problems of object discovery and mass estimation, primarily using self-supervised learning mechanisms, including interaction.

**Segmentation by interaction.** A large number of past works in the robotics community have explored the problem of segmenting objects by interacting with them [18, 29, 6, 41, 59, 24, 44, 7, 14]. These works typically use physical robots requiring them to also overcome low level manipulation challenges. Due to the size and anchored nature of these robots, experiments are typically carried out in constrained laboratory settings, which may not generalize to real world scenarios with complex backgrounds. While our experiments are carried out in simulation, AI2THOR provides a large number of scenes with varied backgrounds and objects. [45] address instance segmentation by interaction, also dealing with physical robots and simple backgrounds. It assumes that objects go out of the field of view after the interaction and the images are captured in a fixed setup by multiple cameras, simplifying ground truth estimation. [2] learn an intuitive model of physics by poking objects and [48] identify grasp locations using self-supervision. Our approach addresses orthogonal problems of discovering objects and learning attributes.

**Object Proposals.** The task of identifying all objects in an image is sometimes referred to as object proposal generation, and is an essential component of many state-of-the-art object detectors

[49, 26]. Both supervised (e.g., [65, 33]) and weakly supervised (e.g., [56]) object proposal generation approaches have been developed. In this paper, we tackle a similar problem, but we rely on interaction and self-supervision to find objects. Past works have explored unsupervised approaches for object proposal generation from single images. [13] propose an approach for finding 3D proposals in RGBD images using geometric means. Unlike our approach they do not use motion cues in an interactive environment. [15] propose a generative model to find objects in an image. However, their approach does not scale to large images or images with multiple objects.

**Multi-frame Object Discovery.** A large body of work addresses the problem of finding objects in multiple images or videos. The supervised video object discovery approaches use different forms of supervision such as CNNs pre-trained for other tasks [35, 58, 57], supervised object proposal models [19, 11, 34], fixation data as annotation [61], or instance segmentation for single frames [46]. Some of these approaches are considered unsupervised in that there is no supervision across the video frames. In contrast, our method is self-supervised and we do not rely on any annotation.

Unsupervised and semi-supervised object discovery approaches have been explored as well [51, 53, 43, 40, 55, 63, 12, 8, 60, 38]. [43, 40, 8] require annotations for the first or few frames of the video. In contrast, our method requires no annotation. [55, 63, 60] require a video at test time to find the moving objects. Our approach infers the objects from a single image. [51] propose an unsupervised approach to discover objects in a collection of web images. In contrast, we identify objects without any prior knowledge of objects in images. [38] propose a multi-granular approach to find objects in videos. [53] find object proposals by sampling a saliency map. [12] also propose an unsupervised approach, which uses PCA reconstruction. Note that all these approaches typically assume the object is moving. This assumption is not valid in general, especially in indoor environments, since the objects are typically at rest. Hence, our agent depends on its actions to set objects in motion.

**Mass Estimation.** Estimating mass from visual data has been explored in different contexts. [62] infer the mass by observing objects sliding down a ramp. [54] infer the mass in a supervised setting. There are also approaches for mass estimation for specific object categories (e.g., [47, 3]). Our work differs from these works since we infer the relative mass of objects by interacting with them.

## 3 Task: Self-Supervised Object Attribute Estimation

Humans typically learn about objects in their vicinity by observing them as well as interacting with them. There is little to no external supervision given by other humans. We present a computational approach to discover objects and estimate their attributes in a self-supervised fashion, using no external supervision.

Our agent does not interact with the physical world, but instead, is instantiated within AI2THOR [30], a virtual near photo-realistic environment with a physics engine, that is often used to study embodied agents [27, 20, 28, 64, 22]. AI2THOR provides a large number of indoor scenes within which agents may navigate around, reach for objects, apply forces to them, pick and throw them. Hence, it is a suitable testbed for our interactive agent to learn about its world. In this work, we attempt to discover objects with no supervision and estimate two attributes - geometric extent (via a 2-d segmentation of the observed pixels) and an appropriate force to move the object, which we refer to as relative mass. At each training episode, the agent is spawned at a random location within one of the scenes, and observes an ego-centric RGB+Depth view. It then actively generates raw visual feedback by picking $N$ points $(u_i, v_i)_{1:N}$ within the image to interact with, each with an interaction force magnitude $(f_i)$. These forces are applied sequentially in time, to the points $(u_i, v_i)$. The force is applied to any object or structure within the reach of the agent along the ray originating from the center of the camera and passing through that point. If the point $(u_i, v_i)$ does not correspond to any movable object or the force pushes an object against static obstacles, there is typically little to no perceived change in the observation[1], but if it corresponds to an object and the chosen force is sufficiently strong, the object moves. The agent then receives raw visual feedback consisting of the RGB view after the scene has come to rest. Through a sequence of interactions within the training scenes, the agent must learn to (1) identify points $P$ in an observation that are likely to cause motion, corresponding to interactable objects and (2) estimate their attributes. During evaluation, the agent must predict these without interacting with the scene, instead using only a single visual observation as input.

Self-supervised object discovery with attribute estimation poses several challenges: (1) *Sparse supervision:* Supervised learning frameworks typically provide dense annotations. For example, instance segmentation datasets provide mask annotations for all object instances within each target category. In contrast, our agent obtains supervision only for points that it chooses to interact with. The remaining points cannot be safely regarded as background since a scene may have unexplored objects within it. The small number of interactions per scene (owing to a fixed budget) and their sequential nature (so that scene changes accumulate over time) prevent the agent from obtaining dense supervision. (2) *Noisy supervision:* In our setting, supervision must be computed from the observations before and after interaction and cannot utilize a trained surrogate model and thus tends to be very noisy. In addition, some of the supervision is inherently ambiguous. For instance, small movements can only help segment an object partially. (3) *Class imbalance:* In a typical household scene, objects that can be moved by a force occupy a tiny fraction of the total room volume. This causes a high imbalance between object and non-object pixels, which complicates learning. (4) *Efficiency:* While significantly cheaper than using a physical robot, interacting with a virtual environment is time consuming, compared to methods dealing with static images. Our agent is assigned a fixed budget of interactions and must learn to use this wisely.

# 4  Model

Our model design is a convolutional neural network inspired by past works in clustering based instance segmentation [42, 16]. As shown in Figure 2, it inputs a single $300 \times 300$ RGB+D image and passes it through a UNet style backbone [50], which also consumes absolute pixel coordinates similar to [37]. The network produces three output tensors (shown in orange in the top portion of Figure 2), each with a $100 \times 100$ spatial extent: (1) An interaction score per location - This signifies the confidence of the model that an interaction with this pixel will result in a change in its observation. (2) Force logits per pixel - These indicate the minimum force magnitude the model predicts will be necessary to achieve such a change. In practice, we quantize force magnitudes into 3 bins, resulting in 3 logits per location. (3) Spatial embeddings - These are computed for each spatial location and capture the appearance of that location. The embeddings are used in a clustering algorithm to compute object instance segmentation masks, which encourages locations within a single object to have similar embeddings, and locations across objects to have different embeddings. Each output is trained with its own loss function. These tensors are used to select actions during training and estimate object attributes at inference.

The network has a field of view large enough to include several objects but small enough to ensure sufficient resolution for small objects within a $300 \times 300$ image. Our network has relatively small number of parameters (1.4M), which helps stabilize training. See the supplement for more details.

## 4.1  Inference

At inference, the model predicts points and force magnitudes of interaction, and a binary instance proposal segmentation mask corresponding to each point, via the following algorithm. During training, the agent uses the same inference procedure to select a set of actions (locations $(u_i, v_i)_{1:N}$ and forces $(f_i)_{1:N}$).

**Given**: A network forward pass, desired number of actions ($N$) and interaction score threshold ($\theta$).
**Iterate** until $N$ actions selected or all interaction scores $< \theta$:

1. Select the pixel $\mathbf{p} = (u, v)$ with the highest interaction score. Select the force magnitude $f^r$ (details in Sec 4.2-II) with $r = \operatorname{argmax}_r m_{\mathbf{p}}^r$, where $(m_{\mathbf{p}}^r)_{r=1:3}$ denotes the vector of force logits at $\mathbf{p}$. Further, denote by $e_{\mathbf{p}}$ the $16D$ embedding vector of this pixel.
2. Add the pixel $\mathbf{p}$ and $f^r$ to the list $(u_i, v_i)_{1:N}$ of action points and $(f_i)_{1:N}$ of forces.
3. Consider the set $\tilde{P}$ of pixels $\mathbf{p}'$ such that[2] $\|e_{\mathbf{p}'} - e_{\mathbf{p}}\|_2 < 1$. Define $e$ to be the $16D$ mean vector of all $e_{\mathbf{p}'}$, $\mathbf{p}' \in \tilde{P}$.
4. Consider the set $P_M$ of pixels $\mathbf{p}'$ such that $\|e_{\mathbf{p}'} - e\|_2 < 1$. Add this set to the list of proposed object segmentation masks.
5. Set interaction scores and embedding vectors to $-\infty$ for all $\mathbf{p}' \in P_M$.

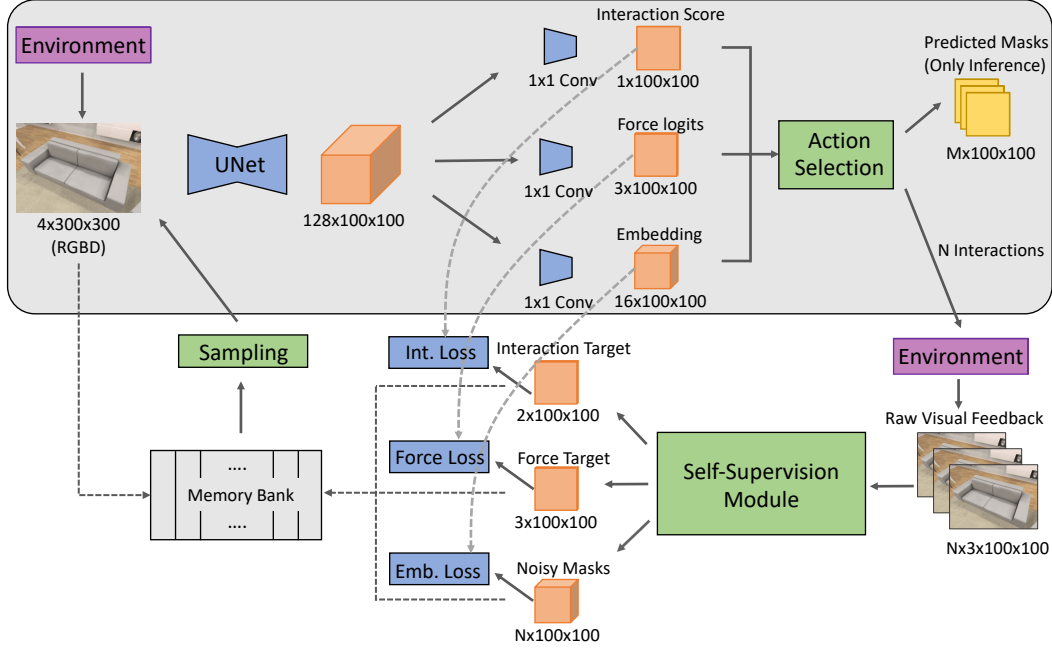

Figure 2: **Model overview**. Our model is a convolutional neural network that receives an RGB+D input and outputs instance masks and relative mass estimates. The grey box shows the portion of the network used during inference. The outputs of each network component are shown in orange. We compute three types of losses: Interaction loss, Force loss and Embedding loss. Note that both the predictions and, via interaction, the ground truth are generated by the network.

Note that the relative mass (predicted force magnitude) of an object is the argmax of the force logits at the pixel that the model chose for interaction with the object. While we predict such force logits for every pixel in the image, they are used only on points selected for interaction.

Intuitively, each iteration in the algorithm above runs one step of an EM clustering algorithm with seed selected by the interaction score, and an $L^2$ based similarity metric for the embedding vectors. Step 1 chooses the maximum interaction score among all pixels that have not yet been clustered as part of an object, and so the algorithm is greedy. During self-supervised training, we further add random actions to the actions selected by the algorithm, to aid exploration.

### 4.2 Training

Training the network using only self supervision proceeds iteratively as follows:

**I: Action selection.** The agent is spawned at a random location within the training scenes. The model receives an RGB+D image $I_1$ as its observation, passes it through the network and selects interaction locations and corresponding forces. This is identical to the inference procedure outlined above.

**II: Interaction.** The agent obtains a sequence of raw visual feedback by applying forces to the chosen interaction points in the order determined by the model. Let $F = \{f^0, f^1, f^2\}$ be a set of force magnitudes that are chosen *a priori* and correspond to magnitudes typically required to move light, medium and heavy objects[3]. Let $D$ be a set of 8 force directions corresponding to a quantized set of directions in space, but avoiding those that point into the ground. Finally, let $r$ be the predicted relative mass of the object (if any) at the interaction point ($r = 0, 1, 2 \rightarrow$ light, medium, heavy).

To test whether the chosen force magnitude $f^r$ is too large for the object proposal, the agent first applies a force with magnitude $f^{r-1}$ to the chosen point, with force direction randomly drawn from $D$. If no change is detected in the visual feedback, it increases the force to the predicted magnitude

$f^r$. If again no changes occur, it applies the maximum force magnitude ($f^2$ in our case), which by definition will move any un-obstructed, interactable object if one were present at the chosen point. This provides noisy supervision that helps the agent learn whether the predicted force was too large, just right, or too low. Note that the agent might apply a force in an undesirable direction resulting in no change (e.g., pushing an object against a wall or other obstacles). Such issues introduce further noise into the supervision signal, but can't be avoided in realistic scenarios.

**III: Self supervision.** Our goal is to only use self-supervisory signals to train the network. Hence, we rely on simple visual changes between the observations, prior to and after the application of the force. We detect changes in the scene using image difference. Consider a pair $I_k$, $I_{k+1}$ of consecutive views of the scene, before and after interaction, transformed to HSV space. We downsample the image difference $J = I_k - I_{k+1}$ using mean pooling to have the same size as the output of the forward pass. We compute a binary mask $B = \mathbb{1}(J^2 > 0.01)$, where $J^2$ is the pixel-wise $L^2$ norm.

To alleviate some of the noise introduced into the supervision and to align the self-supervised mask along the edges of objects, we employ an unsupervised low-level grouping of pixels to form superpixels [17] and use them to post-process the mask $B$ into a more robust mask $B^+$, which is a union of all superpixels that overlap sufficiently with the noisy mask $B$. More precisely, we compute superpixels for image $I_1$ (the original image received by the model). For each superpixel, if at least 25% of its pixels belong to $B$, we add the superpixel to the new mask $B^+$. If the number of pixels on $B^+$ in the immediate vicinity of the interaction location, weighted inversely by distance, is less than a threshold, we declare that the scene has not changed and consider the interaction "unsuccessful". Otherwise, we consider it "successful", and consider $B^+$ as the (noisy) segmentation mask of the object that moved during the interaction. While this method is simple and completely unsupervised, it is noisy due to imperfect thresholding, appearance challenges such as change in shadows, movements of multiple objects via a single interaction, object state changes (e.g., a high force might break a glass), partial overlap between object masks in the two frames, etc.

Finally, the initial frame $I_1$, successful interaction points, and associated supervision masks $B^+$, predicted force magnitudes, and the true force magnitudes that produced the change for these interactions, as well as the points leading to unsuccessful interactions are added to a memory bank.

**IV: Learning from the memory bank.** After a number of interactions, gradient updates are performed on batches sampled from the memory bank based on an importance score. We have noticed in experiments that under-sampling medium-loss training data yields substantial performance gains, possibly because fitting the noise patterns in such data would be detrimental to the learning process. This should be compared to prioritized replay methods [52] (where high-loss, difficult, images are over-sampled) and self-paced learning [32, 5] (where low-loss, easy images are over-sampled). More precisely, data points that are yet to take part in an update are assigned a high importance score and data points with no detected objects (in step III) are assigned a low score. When a data point takes part in an update, its score is reassigned based on the IoU of its predictions with the noisy segmentation targets of the memory bank. Details are in the supplementary.

For each batch of image-target pairs added to the memory bank, we sample $K$ batches from the bank and backpropagate the losses on these batches. $K$ is annealed during training. Interacting with a simulated world, while not as slow as a real robot, still consumes a fair bit of time due to invoking the physics engine. A memory bank allows every interaction to be sampled multiple times over training, making our model more robust and also more efficient than a pure online approach. We use a fixed size memory bank (20,000 points). Once filled, new points replace the oldest ones. After gradient updates for each batch, the agent is spawned at new locations and repeats this process.

## 4.3 Loss Functions

We now describe the loss functions used to learn the interaction score, force logits, and the pixel-wise spatial embedding vectors. Recall that for each scene the agent has previously interacted with, the memory bank contains successful and unsuccessful locations of interaction, and also force magnitudes and noisy masks for each successful interaction.

**Interaction score loss.** We want to yield high interaction scores at the points of successful interactions, low scores at points of unsuccessful interactions, but do not supply the model with gradients from unexplored regions of the image (where no interactions were performed). We first form two targets: a binary map for foreground locations and one for background locations. These are smoothed

|  | Bounding Box | | Mask | |
|  | $AP^{IOU=0.5}$ | AP | $AP^{IOU=0.5}$ | AP |
| --- | --- | --- | --- | --- |
| **Joint segmentation & mass:** | | | | |
| (a) Ours (self supervision) | 24.19 / 27.59 | 11.65 / 13.44 | 22.00 / 25.01 | 10.24 / 11.00 |
| (b) Mask-RCNN [26] (full supervision) | 33.94 / 50.72 | 22.08 / 34.44 | 28.21 / 45.57 | 16.75 / 30.12 |
| **Segmentation only:** | | | | |
| (c) Ours (self supervision) | 27.06 / 26.10 | 12.71 / 12.05 | 24.37 / 23.70 | 11.09 / 10.30 |
| (d) Ours w/ supervised masks | 30.77 / 38.44 | 16.83 / 21.25 | 27.46 / 36.49 | 13.89 / 18.67 |
| (e) Ours w/ supervised interaction | 26.11 / 28.06 | 12.26 / 13.23 | 24.87 / 26.14 | 11.53 / 11.71 |
| (f) Ours (full supervision) | 30.23 / 25.68 | 14.86 / 12.03 | 27.40 / 24.54 | 13.23 / 11.00 |
| (g) VideoPCA [12] (supervised interaction) | 8.10 / 7.48 | 3.99 / 3.45 | 7.33 / 6.97 | 3.27 / 2.82 |
| (h) Mask-RCNN [26] (full supervision) | 36.49 / 49.86 | 22.92 / 31.63 | 30.66 / 44.87 | 18.35 / 28.85 |
| (i) Mask-RCNN [26] (self-supervised masks) | 15.93 / 21.26 | 8.16 / 11.39 | 10.88 / 15.84 | 6.08 / 8.73 |
| (j) Robust Set Loss [45] (self-supervised masks) | 17.51 / 22.29 | 9.11 / 12.08 | 10.71 / 16.23 | 4.02 / 8.81 |
| *Ablations (segmentation only)* | | | | |
| (k) Ours w/o superpixels | 12.74 / 13.24 | 4.77 / 4.93 | 10.21 / 10.72 | 3.81 / 3.75 |
| (l) Ours w/o prioritized sampling | 21.04 / 21.32 | 8.70 / 8.78 | 19.02 / 19.68 | 7.75 / 7.76 |
| (m) Ours $\infty$ arm length | 21.06 / 20.10 | 9.38 / 9.36 | 20.64 / 17.26 | 9.15 / 7.56 |

Table 1: **Object segmentation.** Results are for *NovelObjects* / *NovelSpaces* scenarios.

via a Gaussian kernel and then used in a KL divergence-based loss, but with gradients that decay faster with model confidence, similar to a focal loss [36].

**Force loss.** For each successful interaction, the memory bank stores if the predicted force magnitude $f^r$ was (1) correct, (2) too small to move the object, or (3) too large. The force loss penalizes cases 2 and 3. In case 1, we want to increase the $r$th force logit at this position. In case 2, we provide gradients that increase all force logits $s$ with $s > r$. In case 3, gradients increase all force logits $s$ with $s < r$. Pixel-wise force logits only receive gradients at successful interaction locations.

**Embedding loss.** The inference method outlined in Sec 4.1 employs a clustering method over embedding vectors to determine instance segments. Our goal is to decrease the variance among embedding vectors that belong to the same object and simultaneously increase the dissimilarity of embedding vectors belonging to different objects. We use the noisy masks produced by the self-supervision module to supervise the embedding vectors. For each mask, we compute the mean embedding vector $e$ of all pixels on the mask, and compute $\|e_{\mathbf{p}} - e\|_2$ for all pixels. We provide gradients that decrease this distance for all $e_{\mathbf{p}}$ with $\mathbf{p}$ belonging to the mask, and increase the distance for all $\mathbf{p}$ not on the mask. To stabilize training, inspired by the focal loss, our loss has gradients that decrease rapidly for confident predictions (very large or very small $\|e_{\mathbf{p}} - e\|_2$).

The primary advantage of these losses, and a clustering approach to segmentation, is that no gradients are associated to unexplored regions of the image, addressing one of our key challenges above. The losses are robust to noise, and sparse enough to be partially shielded from class imbalance between object and non-object pixels, addressing the other important challenges.

## 5  Experiments

We now describe experiments and ablations on our self-supervised method for instance segmentation and mass estimation. As a point of reference, we also include results for different levels of supervision.

**Framework.** Objects and interactions in AI2THOR [30] are governed by an underlying Unity physics engine. This enables objects to have fairly realistic physical attributes like mass, material and friction. AI2THOR contains four types of scenes: *kitchens*, *living rooms*, *bedrooms*, and *bathrooms*, with 30 rooms of each type. We conduct experiments with two data splits: (1) *NovelSpaces* - We use 80 scenes for training (20 in each type) and report results on 20 different rooms (5 in each type). (2) *NovelObjects* - We use kitchens and living rooms in train (60 scenes) and bedrooms (30 scenes) in test. In this scenario, the majority of object categories encountered in test are novel (e.g. pillows are only seen in bedrooms). In addition, object locations within each scene in AI2THOR are also randomized. This provides ample variability during training.

|                              | Mean per Class Acc. | $AP^{IOU=0.5}$ (Mass & BBox) |
|------------------------------|---------------------|------------------------------|
| Ours (self-supervised)       | 50.79 / 55.86       | 11.85 / 11.01                |
| Mask-RCNN (full supervision) | 78.28 / 86.11       | 26.90 / 46.24                |

Table 2: **Mass estimation.** Results are for *NovelObjects / NovelSpaces* scenarios.

In total, there are 17,211 locations with 2,765 interactable object instances within the reach of the agent split across scenes in both data splits. AI2THOR includes 125 object categories in total, where 86 of them are interactable. Note that, in many locations, there are no objects in reach of the agent to interact with, and it should learn no interactable objects exist in those locations.
We train all networks from scratch. Implementation details are provided in the supplementary.

**Object segmentation.** Table 1 shows results for segmenting objects. We report standard COCO metrics [1] for detection and instance segmentation. Row (a) shows results for a model that predicts segments and mass jointly. This shows fairly strong results, given no annotations during training, even in the harder NovelObjects scenario; illustrating our approach generalizes well to object categories that have never been seen during training. (c) shows results for a model trained only for segmentation. It is interesting that in spite of solving a harder task, (a)'s metrics do not reduce a lot compared to (c).

**Mass estimation.** We evaluate the model for mass estimation using three buckets: mass $m < 0.5$ kg; $0.5 \leq m < 2$ kg; and $m \geq 2$ kg with results in Table 2. The reported metric is mean per class accuracy, for which chance is at 33%. We also report AP in which case a bounding box is considered correct if both the extent and the mass is predicted correctly. This is a strict metric and, as can be expected, the model shows low scores. For self-supervision, we fix the set of forces to $f^0 = 5N$, $f^1 = 30N$ and $f^2 = 200N$ in order to provide reasonable interactions with objects in each bucket.

**Providing more supervision.** While our method has been developed for self-supervised learning from noisy feedback, we also explore how supplying ground truth information aides the learning process. For these, we consider only the pure instance segmentation model within Table 1. (e)-*Supervised interactions* shows improved results for NovelSpaces, when an oracle supplements interaction locations for all objects in the scene not selected by the model. (d)-*Supervised masks* shows results when an oracle provides the true segmentation masks of all objects selected for interaction by the model. This shows a huge improvement over (c), reflecting the difficulty of learning from noisy masks in a self-supervised setting. Finally, (f)-*Full supervision* combines both levels of supervision. Surprisingly, this does not perform much better than (c). Qualitatively, we notice that (f) over-segments objects and is usually over-confident about them. While this behavior may be due to the fact that our model and training design have been developed for the noisy self-supervised case, we also conjecture that self-supervision and noise tend to focus the attention of learning on "easier" objects (such as ones that move more consistently and noise-free when interacted with), which provides a natural pace for the learning progress of the model.

As a reference, (h) reports the performance of a ResNet-18 based Mask-RCNN [26] with RGB+D input trained with standard, full supervision. We use all standard settings such as optimizers, learning rate schedules, etc. Since the ResNet-18 has about 7 times more parameters than our backbone, this precludes a direct comparison to (f), but provides a useful point of reference. In (i) and (j), we report the performance of this model trained on self-supervised masks obtained from "oracle" interaction as described above, using pixel-wise cross-entropy loss in (i) and the robust set loss from [45] in (j). In [45], this loss was proposed to deal with noisy segmentation masks in their self-supervised learning scenario. The performance of both losses in our case is similar. The massive drop from (h) once again indicates the difficulty of learning from noisy masks.

**Other types of self supervision.** We also provide the result of [12] which extracts segmentation masks using a principal component analysis of the observation sequence as a result of the agent's interaction (row (g)). This method was not effective in discovering objects. We also used an unsupervised optical flow method [39] to compute supervision masks, which did not work well either. These results highlight the difficulty of signal extraction from raw visual feedback.

**Ablations.** We first ablate the effect of the visual prior built into our self-supervision module - using super-pixels as a post-processing step to obtain supervision masks (row (k)), resulting in a large drop. Next, we ablate the effect of our memory bank sampling procedure (row (l)), which also shows a large

| Observation | Interaction score | Mass prediction | Instance prediction | Groundtruth | Self-supervised GT |
|---|---|---|---|---|---|

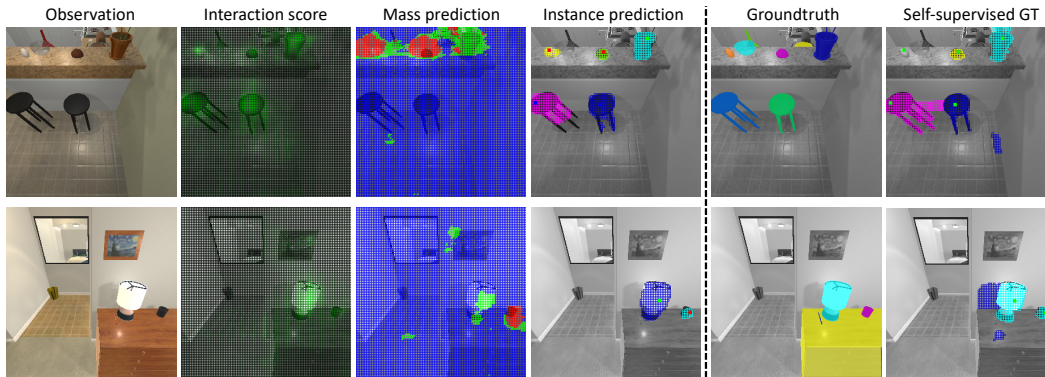

Figure 3: **Qualitative results.** Instance segmentation and mass prediction results are illustrated. The brighter green corresponds to higher interaction score. The masses are shown in red (light), green (medium) and blue (heavy). The selected interaction points are shown with colored dots.

drop, validating our design choices. Row (m) shows the effect of increasing the agent's interaction radius; increasing the number of objects used for evaluation, and rendering the task harder.

**Qualitative results.** Figure 3 shows some qualitative results of our self-supervised approach used to estimate object extents (instance segmentation) and masses. Our model can recognize multiple objects even in cluttered scenes, and for many object types produces relatively accurate masks. The method has difficulty detecting tiny objects such as pencils or forks (which need high precision for successful interaction) and large objects such as sofas or drawers (which move only slightly during interaction, and whose movement is usually detected only at the boundary of the object). Predicting masses from visual cues is expectedly a hard problem as seen by our quantitative and qualitative results, but our model does reasonably well to distinguish 'light' and 'medium' from 'heavy' objects.

# 6 Conclusion

An important component in visual learning and reasoning is the ability to learn from interaction with the world. This is in contrast to the most popular approaches to the computational models of vision that rely on highly curated datasets with extensive annotations. In this paper, we present an agent that learns to locate objects and predict their geometric extents and relative masses merely by interacting with its environment. Our experiments show that, in fact, our model obtains promising results in estimating these object attributes without any external annotation even for object categories that are novel and not observed before. Our future work involves inferring more complex object attributes such as different states of objects, friction forces and material properties.

## Broader Impact

This paper promotes the idea of *learning from interaction* based on self supervision. In recent years, large-scale datasets have led to significant advancements in core computer vision problems. However, curating these datasets is a costly and time-consuming process. This paper is a step towards learning like humans, which typically happens by interacting with the surrounding world instead of supervised training with massive datasets. The broader impact of this research is to show that promising results can be obtained via this method of supervision, and to encourage our colleagues in the community to pursue this direction. We do not expect such methods of learning to have short or long term negative consequences. However, we caution that learning from interaction using physical robots in the real world may have safety implications for other agents and objects in the scene. For this reason, we recommend carrying out real-world studies in constrained laboratory settings or using simulated environments in the near future.

**Funding disclosure.** This work was supported by the Allen Institute for AI.

## Footnotes

[1]Small changes may be observed due to fluctuations in lighting.

[2]Threshold 1 is determined by the scale of the Embedding Loss. See supplementary for details.

[3]We have selected typical minimal forces that move objects which correlate strongly, but not perfectly, with true physical masses. Our model learns such forces, and we evaluate against (quantized) ground truth masses.

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
