[Supplementary Material]

# Learning About Objects
# by Learning to Interact with Them
# –Supplementary Material–

## 1 Model details

Our backbone begins with a $5 \times 5$ convolution with stride 3 and 32 channels. All convolutions except for the final ones are followed by BatchNorm and ReLU, which we will assume implicitly from now on. The initial convolution is followed by three convolutional blocks of stride 2 with 64, 128 and 256 output channels, respectively. Each block consists of two $3 \times 3$ convolutions with a $1 \times 1$ convolution in between, and a residual connection between the input of the block and the input of the last convolution.

The blocks are followed by three transposed-convolutional modules, with lateral connections to the inputs of the block, inspired by the original UNet. Each of the modules applies a $2 \times 2$ transposed convolution with stride 2 to the output of the corresponding block, concatenates the result with the input to the block, and then applies a $3 \times 3$ convolution to obtain a tensor of the same shape as the input to the block. An exception is the final transpose-convolutional module, where we increase the number of output channels to 64.

The resulting $64 \times 100 \times 100$ tensor is concatenated with absolute pixel coordinates, and $1 \times 1$ convolved to obtain the final output of the backbone, a tensor of size $128 \times 100 \times 100$.

The computation of each 128 dimensional entry in this tensor depends on the input inside a $137 \times 137$ box (at the original $300 \times 300$ resolution). This relatively small receptive field is still large enough to fit the vast majority of objects in our dataset, and, as typical for UNet-like architectures, our model output also has direct access to intermediate features computed at high resolution, enabling precise localization of features.

The prediction heads are simple $1 \times 1$ convolutions, with number of channels equal to 1 for interaction scores, 3 for force logits, and 16 for the embedding vectors. In total, the model has 1.4M parameters.

## 2 Loss details

**Interaction score loss.** Denote by $\mathsf{fg}_{\mathbf{p}}$, $\mathbf{p} \in [0, \ldots, 100]^2$, the *foreground*, constructed by placing 1s at positions $\mathbf{p}$ of successful interactions into an initially empty mask, and then convoluting with the $5 \times 5$ filter

$$\mathsf{kernel} = \left(e^{-u^2 - v^2}\right)_{u,v \in [-2:2]}. \tag{1}$$

Similarly, denote by $\mathsf{bg}_{\mathbf{p}}$ the *background*, constructed in the same way from the unsuccessful interaction locations.

Let $s_{\mathbf{p}}$ be the pixel-wise interaction score. Then, our *loss gradient* for $s_{\mathbf{p}}$ is given by

$$\mathsf{grad}_{s_{\mathbf{p}}} = \mathsf{fg}_{\mathbf{p}} \cdot \sigma(-s_{\mathbf{p}}) \cdot e^{-\frac{1}{2}(\max(s_{\mathbf{p}}, 0))^2} - \mathsf{bg}_{\mathbf{p}} \cdot \sigma(s_{\mathbf{p}}) \cdot e^{-\frac{1}{2}(\min(s_{\mathbf{p}}, 0))^2}, \tag{2}$$

where $\sigma$ is the sigmoid function. Our loss function is the integral of that gradient (only the gradient itself is needed for training). In the absence of the exponential factors in $\mathsf{grad}_{s_{\mathbf{p}}}$, this would be a Kullback-Leibler divergence type loss. The exponential factors ensure that confident scores give rise

to very small gradients, like in the focal loss, but our suppression of such gradients is more aggressive than for the standard focal loss.

**Force loss.** There is a force magnitude $r$ ($r = 0, 1, 2$) associated to each successful interaction (namely the magnitude predicted at the time of interaction), and feedback reflecting whether this force was: 1. just right, 2. too small, or 3. too large. Form the $3 \times 100 \times 100$ tensor $(\tilde{\mathsf{ft}}_{\mathbf{p}}^{r})_{r,\mathbf{p}}$ that is nonzero only at pixels $\mathbf{p}$ of successful interactions, and such that, if $\mathbf{p}$ is a successful pixel with predicted force $r$, we have

$$\tilde{\mathsf{ft}}_{\mathbf{p}}^{r'} = \begin{cases} \mathbb{1}(r' = r) & \text{case 1} \\ \mathbb{1}(r' < r) & \text{case 2} \\ \mathbb{1}(r' > r) & \text{case 3} \end{cases} \tag{3}$$

Next, normalize $\tilde{\mathsf{ft}}$ to have mean zero over its first dimension (the one of size 3), and $L^1$ norm equal to 1 over this dimension wherever it is nonzero. Finally, convolve the result with kernel from the previous paragraph to obtain $(\mathsf{ft}_{\mathbf{p}}^{r})_{r,\mathbf{p}}$, the *force targets* for the predicted pixel-wise force logits $(m_{\mathbf{p}}^{r})_{r,\mathbf{p}}$.

Using this notation, the loss gradient for the force logits $(m_{\mathbf{p}}^{r})_{r,\mathbf{p}}$ is given by

$$\mathsf{grad}_{m_{\mathbf{p}}^{r}} = \mathsf{ft}_{\mathbf{p}}^{r} \cdot \left( \mathbb{1}\big[\mathsf{ft}_{\mathbf{p}}^{r} > 0\big] \cdot \sigma(-m_{\mathbf{p}}^{r}) \cdot e^{-\frac{1}{2}(\max(m_{\mathbf{p}}^{r},0))^2} + \mathbb{1}\big[\mathsf{ft}_{\mathbf{p}}^{r} < 0\big] \cdot \sigma(m_{\mathbf{p}}^{r}) \cdot e^{-\frac{1}{2}(\min(m_{\mathbf{p}}^{r},0))^2} \right). \tag{4}$$

This formula is similar to the one used in the loss of the interaction score and was inspired by the same considerations. We notice considerable noise in the gradients for the force logits, and we address this by adjusting the gradients $\mathsf{grad}_{m_{\mathbf{p}}^{r}}$ by a factor of 0.1 relative to those of the interaction score and embedding loss.

**Embedding loss.** A loss gradient is associated to each noisy segmentation mask corresponding to a successful interaction of the agent. The gradients for each such mask are weighted inversely to the area (number of pixels) of the mask and then added, which defines the full loss gradient for the embedding vectors.

We now define the loss gradient for a single noisy segmentation mask $B^+$. We denote by $e_{\mathbf{p}}$ the $16 \times 100 \times 100$ tensor of pixel embedding vectors, and introduce the mean embedding vector $e = \frac{1}{|B^+|} \sum_{\mathbf{p} \in B^+} e_{\mathbf{p}}$.

We compute the $100 \times 100$ tensor $d_{\mathbf{p}} = \mathsf{huber\_square}(e_{\mathbf{p}} - e)$, where $\mathsf{huber\_square}(x)$ computes the $16D$ squared $L^2$ norm in the forward pass, but in the backward pass truncates gradients to have absolute value less than 1 (we want to avoid larger gradients for stability of training). Loss gradients for $d_{\mathbf{p}}$ are given by the following formula, and define gradients for the embedding vectors $e_{\mathbf{p}}$ by backpropagation:

$$\mathsf{grad}_{d_{\mathbf{p}}} = \mathbb{1}(\mathbf{p} \notin B^+) \cdot e^{-(d_{\mathbf{p}}/1.5)^4} - \mathbb{1}(\mathbf{p} \in B^+) \cdot \frac{1.5 \cdot d_{\mathbf{p}}}{1 + d_{\mathbf{p}}}. \tag{5}$$

Note that this gradient increases the distances $d_{\mathbf{p}}$ to the mean vector $e$ for pixels $\mathbf{p} \notin B^+$, but decreases the distances for $\mathbf{p} \in B^+$. The gradient magnitude decreases for confident predictions ($\mathbf{p} \in B^+$ and $d_{\mathbf{p}}$ close to zero, or $\mathbf{p} \notin B^+$ and $d_{\mathbf{p}}$ large). Finally, the factor 1.5 sets a scale for the distances $d_{\mathbf{p}}$: We have

$$e^{-(d_{\mathbf{p}}/1.5)^4} \approx \frac{1.5 \cdot d_{\mathbf{p}}}{1 + d_{\mathbf{p}}} \qquad \text{at} \qquad d_{\mathbf{p}} = 1. \tag{6}$$

In other words, the gradient magnitudes for positive (on mask) and negative (off mask) pixels balance when the distance to mean vector $e$ of such a pixel is equal to 1. This explains why we use threshold $\|e_{\mathbf{p}} - e\|_2 = 1$ in the clustering algorithm. The reasoning is the same as for binary cross entropy, where we threshold logits $x$ at 0, the point where the gradient for the positive class, $\sigma(-x)$, equals $\sigma(x)$, the gradient for the negative class.

Note that, if perfect convergence could be achieved during training, the choice of the threshold would be irrelevant (namely, $\|e_{\mathbf{p}} - e\|_2$ would be equal to zero for $\mathbf{p} \in B^+$ and infinity for $\mathbf{p} \notin B^+$). Because of noise and limited capacity of the model, convergence cannot be achieved, and indeed, we observe that threshold 1 in the clustering algorithm usually gives the best performance.

# 3 Priorities in the memory bank

Every image in the memory bank has an associated *priority*, and we sample from the memory bank by first normalizing the priorities across the bank and then sampling without replacement from the resulting probability distribution. A new image is added with priority equal to $0.5$. Its priority gets updated each time it is used in a forward pass, as described below:

We associate a mask pred_mask predicted by the current model to each noisy segmentation mask $B^+$ on the image, as follows: Compute the distances $d_\mathbf{p}$ as in Section 2 above, and define pred_mask $= d_\mathbf{p} < 1$ (a binary mask). Now, compute the IoU of $B^+$ with pred_mask. We define the *score* of the image to be the minimum of these IoUs for all noisy instance masks $B^+$ in the image. In case the image contains no instances, we set the score equal to $0.5$.

The new priority of the image is computed from its score by the formula

$$\mathsf{priority}(\mathsf{score}) = (\mathsf{score} - 0.5)^2 + 0.02. \tag{7}$$

The priority is thus minimal (although nonzero, thanks to the offset $0.02$) for scores around $0.5$, and maximal for either low or high scores, corresponding to small or large minimal IoUs, respectively.

Although we do not claim that more intuitive strategies to choose priorities, like only prioritizing images with small IoUs (following the prioritized replay philosophy), or only those with large IoUs (following certain heuristics in the area of self-paced learning), could not be equally beneficial as the strategy chosen here, we were unable to successfully apply such intuitive strategies in preliminary experiments. We have observed qualitatively that the IoU of noisy instance masks with the *ground truth* mask of the corresponding object is often around $0.5$, and based on this observation, it could be conjectured that sampling images with such noisy masks less often exposes the model to a cleaner signal for instance segmentation.

Note that we do not perform a bias correction of loss gradients, as is sometimes done to ensure the loss gradient remains unbiased despite non-uniform sampling. We have found this to be detrimental to learning in our case.

# 4 Implementation details

**Dataset and interaction.** To initialize a scene for our agent, the AI2THOR controller requires the position (2 coordinates) and direction of view (2 angles) of the agent within the scene, as well as a random seed for spawning objects into the scene. We choose these parameters from a fixed dataset in order to reduce some of the randomness in training.

The agent interacts with a scene only by applying forces, and its maximum radius of interaction is limited to 1.5 meters. All moveable objects whose ground truth segmentation mask includes at least 10 pixels that are within this distance of the agent are included in the ground truth.

In AI2THOR, forces of a given magnitude and direction are applied to the selected point for 1 millisecond, and induce realistic rigid body motion. Points are selected by emitting a ray from the camera through the selected pixel. The first point hit by this ray and corresponding to an object or structure is selected, unless it is out of reach of the agent (farther than 1.5 meters from the camera).

**Training details.** Our network is randomly initialized, and first trained on the instance segmentation task only. We start by filling the memory bank with interactions from 3000 locations of the train set. We then alternate between adding batches from 70 locations to the memory bank, and training on $K$ batches of size 64 sampled from the memory bank, where $K$ is linearly annealed from 15 to 45 over the course of the training. At each location, the model performs $N = 10$ "greedy" interaction attempts according to the algorithm of Section 4.1 of the main text, and an additional 10 random interactions. Our optimizer is Adam with weight decay $10^{-4}$ and learning rate $5 \cdot 10^{-4}$. In total, we perform interactions at approximately 65k locations, and roughly 25k gradient steps.

After this segmentation-only pre-training, we train segmentation and relative mass prediction jointly. We use essentially the same setting as before, but anneal $K$ from 15 to 35, and only perform 5 "greedy" and 5 random interaction attempts at each location.

**Hardware details.** In the setting above, our model can train on a single GeForce RTX 2080 Ti. Interaction with the environment can be parallelized, and needs additional GPU and especially CPU

resources. On our 36 CPU Intel Core i9-7980XE machine, we achieved good performance by running 35 agents in parallel, which require an additional RTX 2080 GPU to simulate physics. In total, training takes about 36 hours in this setup. For inference, our current implementation processes $300 \times 300$ RGB+D images at about 20-25 fps on a single GPU.

**Training details – Mask-RCNN.** Our Mask-RCNN references are trained in detectron2 [3] in two settings: fully supervised (all model inputs pre-rendered, and learning targets use full ground truth information), and instance segmentation with oracle interactions (target segmentation masks are replaced by the output of our self-supervision module). In the latter case, data collection is still independent of the model, and is done prior to training. Training itself can then follow the supervised paradigm (but with noisy targets).

The model setup follows a standard configuration for Mask-RCNN with a ResNet-18 backbone network (all defaults as in detectron2), but with RGB+D input channels. Networks are randomly initialized and trained from scratch. We use a base learning rate of $2 \cdot 10^{-3}$ and reduce it by a factor $10^{-1}$ after 60,000 and 80,000 steps, before terminating at 90,000 steps.

**Providing more supervision.** We describe the interaction and mask oracles used to study the effects of partial supervision on the model performance.

The interaction oracle first filters the actions predicted by the model to contain at most one point on each moveable object that is in reach. It then adds one such point for each object that was not selected for interaction by the model. Other selected points are not affected by the oracle, so that the model can produce its own hard negatives. This modified list of interaction points is then passed on to the agent for interaction.

For each interaction point (2-D), the mask oracle retrieves the ground-truth segmentation mask that point belongs to, or returns the empty mask if the point either does not belong to an object or corresponds to a world location not in reach of the agent. These masks are down-sampled and substituted for the ones usually produced by interaction and the self-supervision module. The rest of the training remains identical to the self-supervised case.

**The videoPCA baseline.** The method [1] trains a foreground-background segmentation model via $L^2$ loss to a soft, noisy segmentation mask extracted from videos. At inference, the foreground is clustered into connected components (instances), and post-processed. We adapted this approach to be applicable to our dataset in the following way:

The video is obtained from AI2THOR by performing oracle interactions as described above. These videos are much simpler than the ones for which the original approach of [1] was designed, and we use less (namely, 4) PCA components for frame reconstruction. Also, moving objects are often not in the center of the image. We replace the centering heuristic of [1] by centering around the hard masks produced in the first step of their approach. Soft masks corresponding to different interactions are added and clipped at 1 to give a single soft mask target. Finally, we adapt the sizes of filters they use for smoothing to the resolution of our videos.

The model we use has the same backbone as our self-supervised model, and is trained for the same number of steps and with the same learning rate, but using $L^2$ loss to a single soft mask score, as in [1]. For post-processing, we use superpixels (rather than a CRF, as in the original).

**Optical flow.** Unsupervised optical flow is a principled approach to finding correspondences between images before and after interaction, and thresholding the optical flow in the initial image can ideally provide clean segmentation masks of the moving object. However, learning unsupervised optical flow is not straightforward and itself prone to noise. In preliminary experiments, we used the pretrained (unsupervised) Unflow network from [2] to extract masks in this way, but did not achieve good performance. It is an interesting research direction to train an optical flow network on our self-supervised data from scratch.

**Ablation with $\infty$ arm length.** We have performed various experiments investigating the effect of the interaction radius of the agent on training. An interesting case is infinite interaction range, where the agent can apply forces to all objects in sight, irrespective of their distance to the agent. In Table 1, row (l), of the main text, we report the performance of such an agent.

**Selection of hyperparameters.** The model, losses, and self-supervision module incorporate a large variety of design and hyperparameter choices. Most of these choices were made using domain knowledge, and were not tuned or ablated by evaluation on the ground truth. There are two exceptions:

- We studied four settings for the interaction range of the agent. In AI2THOR, this range can be a sphere around the camera, a vertical cylinder around the agent, or an intersection of the two. A larger interaction range means more potential signal for the agent to learn from, but also more ground truth objects for the agent to discover. Among the settings we tried, the one we report on (sphere of radius 1.5m) was the second best performing. This is a natural setting (robot arm anchored at a fixed point on the robot).

- We studied five choices for the filter kernel defined in (1), and the threshold 1.5 defined in Section 5 below. We found it to be important that this filter be small and its elements quickly decaying. Especially in its use in the self-supervision module, large filter sizes have a surprisingly detrimental effect. We used the best performing choice in our final setup.

## 5 Self-supervision module details

**Successful vs. unsuccessful interactions.** Every interaction produces a $100 \times 100$ binary mask $B^+$ via thresholding of the difference of images before and after interaction, and alignment with superpixels. Such masks contain various sources of noise, and can be non-empty even if the interaction does not actually move any object in the scene. It is effective to declare interactions unsuccessful if they satisfy the following criterion: restrict the mask to the $5 \times 5$ neighborhood of the point of interaction, and multiply with the filter kernel from equation (1); then, the interaction is unsuccessful if the sum of the entries of the resulting $5 \times 5$ matrix is less than the threshold 1.5. Otherwise, the interaction is successful.

Intuitively, successful interactions thus produce masks that contain at least two or three $3 \times 3$ grid cells (at full resolution) in the immediate neighborhood of the point of interaction (with more weight given to cells close to the point).

## 6 Details on results

**More qualitative examples.** We include some additional qualitative examples in Figure 1. They serve to illustrate the following observations:

- As seen in rows 1 to 3, our model can learn to detect multiple objects of varying shapes, even in relatively cluttered scenes. It does so by learning from a self-supervised ground truth that is sometimes very noisy (as in row 2), and is usually missing objects (like the table and sofa in rows 1 and 3, which are hard to move, and several small objects, which are hard to localize). Relative mass predictions reliably single out heavy from medium or light objects, and the prediction is not just based on the size of the object proposal (e.g. the cardboard box in row 3 is large, but can easily be moved by a medium force).

- Rows 1 to 3 also contain some common failure modes of our model, like over-segmentation, or including objects that are just out of reach (the bottle in the left corner of the images in row 2).

- As seen in row 4, our model sometimes lacks confidence in selecting even easy to move, large objects in uncluttered scenes.

- Conversely, as seen in row 5, our model sometimes confidently selects parts of structures as objects. Note that some of the false positives in that image (especially the kitchen utensils to the left of the stove, which are not moveable and part of a structure in AI2THOR) could well be a moveable object if judged solely by visual clues. Interaction is necessary to learn that they are not.

**The variance of the results.** Since our model creates its own learning signal via self-supervision, final performance of training could be expected to be more sensitive to random seeds than is usually the case in fully supervised training. However, we have not observed large fluctuations as appear sometimes in reinforcement learning. Training is too expensive for us to collect standard deviations

Figure 1: **Additional qualitative results.** (a) Initial observation. (b) Predicted interaction scores. The brighter green corresponds to higher interaction score. (c) Predicted mass. The masses are shown in red (light), green (medium) and blue (heavy). (d) Instance prediction results. The selected interaction points are shown with colored dots. (e) The groundtruth that for objects in the vicinity of the agent. (f) Self-supervised groundtruth, which is noisy and is obtained by interaction. The bottom two rows show failure examples.

| Bounding Box | | Mask | | Mass estimation | |
|---|---|---|---|---|---|
| AP$^{IOU=0.5}$ | AP | AP$^{IOU=0.5}$ | AP | Mean per Class Acc. | AP$^{IOU=0.5}$ (Mass & BBox) |
| $23.77 \pm 0.83$ | $11.33 \pm 0.59$ | $21.92 \pm 0.90$ | $10.01 \pm 0.50$ | $49.79 \pm 1.36$ | $10.75 \pm 1.03$ |
| **Run reported in main text (for reference):** | | | | | |
| 24.19 | 11.65 | 22.00 | 10.24 | 50.79 | 11.85 |

Table 1: **Stochasticity of model performance.** Results are for *NovelObjects* scenarios.

for all results reported in the main paper. Instead, we performed three additional self-supervised training runs on the *NovelObjects* dataset, and report mean and standard deviation among the runs on this dataset in Table 1. This serves to illustrate the fluctuations in performance that can be expected for our self-supervised approach.

**Confusion matrix for mass estimation.** The Mean per Class Accuracy is only one measure to evaluate the performance of our model on the relative mass prediction task. More detail is contained in the confusion matrix of relative mass predictions among successfully detected objects (as in the main text, the BBox-IoU threshold for a successful detection is 0.5). The confusion matrices on both test sets can be found in Table 2. Their rows are normalized. Mean per Class Accuracy is the average of the diagonal entries of the confusion matrix.

|  | *NovelObjects* |  |  | *NovelSpaces* |  |
|---|---|---|---|---|---|
| 0.1318 | 0.5041 | 0.3641 | 0.4193 | 0.5092 | 0.0714 |
| 0.0510 | 0.6114 | 0.3376 | 0.1591 | 0.5413 | 0.2996 |
| 0.0066 | 0.2128 | 0.7805 | 0.0366 | 0.2483 | 0.7151 |

Table 2: **Confusion matrices for mass estimation.** Rows index ground truth classes and columns predicted classes. Indices $0$, $1$, $2$ denote "light", "medium", "heavy". Computed for successful detections in the corresponding test set (with BBox - IoU threshold of $0.5$).