[Reviews · NeurIPS 2020]

Review 1

Summary and Contributions: This paper studies learning object-centric representations from interaction in a self-supervised way. Compared with prior work, the main contribution is that the proposed model also learns physical object properties from interaction and has been evaluated in an embodied environment (AI2THOR).

Strengths: This paper extends earlier work to not only discover objects but estimate their mass and force through self-supervision (interaction). The presentation is clear. The authors included rather comprehensive supp material and code.

Weaknesses: The comparisons are lacking: the authors are comparing with ablated versions of their own model as well as supervised "oracle" models. Beyond those, the authors should also compare with prior models with a similar setup (e.g. [2] and [45]). Technical innovations are limited: the model seems to be a generalized version of [2]. The results are not as impressive: they're all in a simulated environment and the scenes are rather simple. The authors didn't evaluate on down-stream tasks: to me, segmentation and mass are not the end goal of interaction---after objects and masses are discovered from interaction, it'd be great to see how they help in for example manipulation.

Correctness: Yes. This is an empirical paper.

Clarity: Yes, well-written.

Relation to Prior Work: Yes, but comparisons are missing.

Reproducibility: Yes

Additional Feedback:


Review 2

Summary and Contributions: This paper presents a self-supervised agent that learns to localize objects and predict their geometric extent and relative mass by interacting with its environment and without any human supervision. The experiments demonstrate promising results in estimating these attributes for object categories that are both observed as well as unobserved in the training data.

Strengths: - The overall idea and motivation of the work is fresh and convincing. Self-supervised representation learning through interaction in environments makes a lot of sense, and the paper takes a good solid step towards this goal. - The approach predicts the segmentation masks as well as relative mass of objects which is appealing. In particular, predicting relative mass is something that would be extremely difficult to achieve with alternative self-supervised visual representation learning approaches that only learn from visual data. - The experiments overall are thorough and well-conducted. Ablation studies are conducted to demonstrate the effectiveness of each model component, and comparisons to sensible baselines are made.

Weaknesses: - The main weakness of the paper is the technical approach. There are several somewhat arbitrary rule-based decisions including how to merge different superpixels to create a mask (L202); ordered choice of force applications (L187-189); thresholds (L204). A more principled approach with less heuristic design choices would be desirable. Having said that, I understand that this is a very challenging problem setting and do feel that the proposed approach overall is sensible and a good first take.

Correctness: The overall approach is somewhat complex and includes several heuristics (as mentioned above). But the claims and method appear to be correct. The authors have also shared their source code.

Clarity: - It is well-written, but I did find the approach section to be a bit difficult to understand initially, as many details are provided in the supp. - The sampling of the memory bank that combines intuitions from prioritized replay methods and curriculum learning seemed contradictory and not well-explained. Could the authors please elaborate? - In the experiments, the setup for the Mask-RCNN baseline is not entirely clear. In particular, for the novelobjects setting, how is the full-supervised Mask-RCNN trained?

Relation to Prior Work: Yes

Reproducibility: Yes

Additional Feedback: Overall, the paper proposes a convincing problem setting for self-supervised representation learning through interaction in environments. While the approach has some weaknesses, the experiments demonstrate promise of the proposed approach. I like the direction that this paper proposes, so my initial score is accept. Please address the weaknesses and questions mentioned above.


Review 3

Summary and Contributions: This work introduced a self-supervised learning framework for discovering and learning object segmentation masks and mass properties from an agent’s interaction in simulated environments. The key idea is to have a learning agent sample random interactions and apply forces of varying magnitudes to different regions of the environment. Supervision can be derived from the differences in visual observations before and after interaction. Their experimental results demonstrate that such self-supervision from interaction can be used to discover objects and infer physical properties.

Strengths: This work addresses the problem of harnessing interaction and motion to gather additional visual cues for inferring object entities and physical properties which could be challenging or ambiguous from static images. This high-level idea of embodied visual perception is quite appealing to me. The overall research goal pinpoints a promising research direction. The proposed method relies on self-supervision generated autonomously from the agent’s visual experiences. No manual annotation is required. This substantially reduces the cost of scaling up this method to large-scale environments and diverse scenes. The resulting method is capable of learning to predict object segmentations and their mass properties (indirectly represented by the force magnitudes necessary to mobilize them from such self-supervision. The experiments demonstrate that the method can generalize to novel objects and novel scenes.

Weaknesses: - Assumptions and Simplifications While this work tackled an interesting problem of embodied visual perception, it has primarily focused on simulated environments in AI2-THOR, as opposed to prior work, e.g., [45], which has studied a very similar problem of segmentation from interaction on real data. The caveat of using simulated environments is that ground-truth object segmentations and physical properties can be easily accessed, undermining the merits of such interactive perception methods. Furthermore, there doesn’t seem a straightforward extension to make the method amenable to real-world deployment. Several strong assumptions and simplifications of this model might fundamentally limit its potential in real-world environments. Some of the assumptions (correct me if I am wrong) include: No exogenous environment dynamics (the only object movement in the environment is caused by the agent’s interaction); Fixed camera angle (no camera motion before and after interaction); The ability to exert forces of pre-specified magnitudes at arbitrary points of the scene along all directions (no consideration of the limitation of the agent’s embodiment); No consideration of safety and robustness and assume the environment is resettable. Though it is understandable for the authors to use simulation as a proxy for model development while abstracting away the practical real-world constraints, it is worthwhile to include a discussion of these assumptions and simplifications, and ideally provide results to show how relaxing them will make impacts on the self-supervised model’s performances. I believe that the long-lasting impact of this line of work would eventually be in the real world. For this reason, this work could substantially improve its quality by evaluating on real-world video data or on the robotic hardware. - Experiments The major problem of the experimental validation is the lack of interactive perception baselines. The method was compared with fully-supervised Mask-RCNN and a selection of self-baselines for ablation study. However, it omits any prior method that harnessed motion cues for object segmentation and inferring physical properties. Though the authors included a discussion in the related work section, arguing that in their work “our agent depends on its actions to set objects in motion”. This doesn’t seem to be a strictly differentiating factor. In fact, many prior works have used embodied agents in simulation and real-world autonomous robots to generate self-supervised object interactions. And clustering-based motion segmentation of objects from RGB-D data has been extensively studied [Shao et al. RA-L 2018, Xie et al. CVPR 2019]. I believe that it is necessary to include baseline comparisons of these interactive perception algorithms for more convincing evidence of model performance. Table 1: comparison (c) and (e) as well as (d) and (f), it seems that using supervised interactions hurts the performance of segmentation. What is the reason that the additional supervision decreases the prediction quality?

Correctness: Line 282: the authors categorized object masses in three baskets (m<0.5, 0.5<=m<=2, m>2) and claimed that the chance is at 33%, where their mass estimation model achieved a performance around 50-55% accuracy. However, from the qualitative examples shown in the paper, the majority of the mass predictions are blue (heavy). This suggested that the three baskets of object masses are not evenly distributed. What would be the simple baseline of predicting “all heavy” or predicting the majority vote of the class? It doesn’t seem to me that 50-55% accuracy is a lot higher than the chance performance, given the severe discretization of the mass predictions. Line 320: “Predicting masses from visual cues is expectedly a hard problem as seen by our quantitative and qualitative results”. Predicting mass from single images is inherently ambiguous and information incomplete. Do you have any intuition about what the model has learned in order to make mass predictions on novel objects? And any idea how motion cues can be integrated to resolve the ambiguity of predictions on static images.

Clarity: It was not clear how action is selected to generate meaningful interaction with the environment, as the majority of the pixels correspond to static environment fixtures. The authors did point out in Line 132 that class imbalance is indeed an issue as objects that can move only occupy “a tiny fraction of the total room volume”. However, I didn’t see statistics on how many interactions selected by the learning agent were able to generate motions and what are the concrete technical solutions to combat the class imbalance. Is it possible to harness any “objectness” prior, such as from region proposals, to guide the selection of interaction points? How is the method able to discover small objects in the scene? I was confused about the mass estimation evaluation (Line 280-285). Is mass predicted for each pixel or on each object? If the former, how did you evaluate the “mean per class accuracy” in Table 2? And if the latter, how did the model make a single prediction for each object?

Relation to Prior Work: This work falls into the broader category of research work on embodied visual learning and interactive perception. Harnessing interaction and motion to infer object entities and physical properties has been a central research topic of both the computer vision and robotics communities for decades, with a vast amount of prior work on motion-based segmentation and system identification. It is suggested (see the Weaknesses part) that additional baseline comparisons and discussions should be included to highlight the key novelty of this work in presence of relevant literature.

Reproducibility: Yes

Additional Feedback: Line 173, 198: use subscript (rather than superscript) for L2


Review 4

Summary and Contributions: This work proposes to learn about objects (geometric extents and relative masses) by interacting with objects in a simulation platform AI2THOR. This algorithm relies purely on the visual feedback of object movement and uses that as self-supervision. The results demonstrate the effectiveness of this approach compared to fully supervised segmentation algorithms.

Strengths: The idea of this work is very novel, inspired by how infants learn about this world. The algorithm and losses are carefully designed such that the model can learn from noisy self-supervision. Although the numbers do not directly compete with the segmentation model with full supervision, the results are decent and have meaningful outputs, based on the qualitative results. Ablation experiments also show how each supervision affects the final results.

Weaknesses: The concept of pushing per pixel is weird and which might not work if the object is not rigid, e.g. couch. And it's unclear how the forces are applied. L112-114 says the force is along the ray origining from the center of the camera and passing through that point, but L184-185 says the force is chosen randomly from set D. Learning the direction of the forces is definitely interesting, but the agent how need to estimate the depth of each pixel? Minor comment: indexing of r is inconsistent between L164 and L186.

Correctness: The claims and method are correct.

Clarity: The paper is well written and easy to understand

Relation to Prior Work: Related work has a detailed discussion of the difference with previous work.

Reproducibility: Yes

Additional Feedback:

[Author Response · NeurIPS 2020]

We thank all reviewers for their valuable feedback. In particular, the reviewers have supported our:

**research and ideas** (**R4**-promising research direction & appealing, **R5**-novel, **R2**-fresh and convincing),

**experiments** (**R2**-thorough and well-conducted, **R5**-carefully designed) as well as

**presentation** (**R1**-clear, **R2**-well-written, **R5**-well written and easy to understand).

We address all of the concerns below. In particular, we provide *new comparisons* to past works, address the question of performing this *research in simulation* and place this within the context of past works in *interactive perception*.

**R1**: **Segmentation and mass are not the end goal of interaction.** We totally agree. However, addressing more complex tasks requires obtaining reasonable performance on these building blocks, which itself is very challenging in a self-supervised setting. Complex downstream tasks including rich manipulations are part of our ongoing research.

**R1**: **Comparison with [2, 45].** We now provide results for a new baseline [45] - augmenting Mask-RCNN with their Robust Set Loss using their public implementation. Across the suite of metrics, this helps a little beyond Table-1(i), but still inferior to our approach, e.g. 22.3 (theirs) vs 28.06 (ours) on NovelSpaces Box $AP^{0.5}$, row (e). [2] addresses the problem of poking an object from one point to another – very different from us, and not valid to compare to. However, we now provide results for our model trained on their dataset (we do not poke, but use their pokes instead) and with almost no hyperparameter tuning obtain 39.1 for Box $AP^{0.5}$ – validating that our method shows promise on real world data. The higher number (39 vs 28) also indicates the relative complexity of the THOR scenes.

**R1**: **All in a simulated environment and simple scenes** Please see *R4 Research/Assumptions in simulation* below.

**R2**: **Rule-based decisions.** These were made to make the learning process more manageable. Future extensions of this work will address more generic architectures and learning paradigms, to remove some these handcrafted designs.

**R2**: **Prioritized replay and curriculum learning.** Prioritized replay and curriculum learning encourage sampling high-loss and low-loss examples, respectively. Combining these two approaches discourages sampling medium-loss examples. This resulted in high gains for our method.

**R2**: **Fully supervised Mask-RCNN for novel objects.** It is trained with groundtruth mass and segmentation masks (obtained from THOR) for seen objects (no category information is used). Novel categories are not in training images.

**R4**: **Research in simulation.** The end goal of this research is to train interactive agents in the real world. Using a rich, large scale and variable simulator allows us to have faster research iterations via fast training (no mechanical constraints), improve generalization (scene and object variability compared to [2, 45]) whilst ensuring safety. Future work includes (a) deployment on a Locobot robot (b) exploring simulation-to-real transfer via real world fine tuning, along the lines of navigation research in embodied AI (e.g., Habitat ICCV19, CARLA CORL17, iGibson ICRA20).

**R4**: **Assumptions in simulation.** Our setup presently makes some assumptions. However, past real world works also use highly simplified setups. E.g. [45] assumes: (1) objects on a flat surface (2) fixed distance of surface to camera, (3) fairly uniform background, (4) objects going out of the field of view after interaction, (5) fixed camera view, (6) no exogenous motion. Our work relaxes (1)-(4) by providing variable objects, surfaces, rooms and backgrounds.

**R4**: **Interactive perception baselines.** One can cluster past works in this area into two parts: (a) passive methods that use objects motion to segment them (b) active approaches that jointly learn to act and segment. Papers in **a** cannot be thought of as baselines – they are orthogonal to our method. When our agent learns to interact with objects, the resulting motion can be exploited by these works to result in better segmentation. Adding such techniques on top of our proposed method is very interesting and we leave it for future work. Also, papers mentioned by R4 use *supervised* components. Papers in **b** are very relevant, and we provide new comparisons above. Please see response to R1. However, our method is designed to not just provide object masks but also object attributes – an improvement over past algorithms.

**R4**: **Additional supervision decreases the quality.** Without supervision the masks and interaction points are noisy. We believe this serves as a form of data augmentation for our method and helps the model to better generalize.

**R4**: **Majority vote is > 33%.** The metric is mean per class accuracy. So if everything is predicted as the majority class, the accuracy is 100% for that class and 0 for the other 2 classes. Hence, majority vote accuracy is (100+0+0)/3 = 33.3.

**R4**: **Intuition for mass prediction & how to use motion cues.** The model probably learns to associate the size and texture of the objects to the mass. To incorporate motion cues, we can enable our training pipeline during inference. It can predict the interaction points and infer the mass based on the motion caused by interaction.

**R4**: **Class imbalance & Objectness prior.** We circumvent the class imbalance issue by learning the objectness score by focal loss only on pixels selected for interaction (line 238). This improves successful interactions from 4% to 60% during training. We did not use objectness priors (e.g., [56,33]) since they use supervised data.

**R4**: **Mass prediction.** It is per object. We predict an interaction point for objects and a mass map for the entire image. The mass for an object is the mass we predict for the pixel corresponding to the interaction point of that object.

**R5**: **Non-rigid objects & force directions.** Great challenging suggestions and will be considered for future work.

**R5**: **Forces.** L112-4 is about the point that the force is applied to. L184-5 is about the direction of the force vector, which is not necessarily in the direction of the ray from the camera center to the point.

[Meta-Review · NeurIPS 2020]

There was substantial discussion about this paper, and the reviewers are mixed. The major point of discussion was whether the model would generalize to real-world robot situations. All the reviewers agreed the learning component was very interesting and the technical approach is solid, although obscure due to the fact that the supplemental material contains key details. The main concern is that the approach makes assumptions that might only hold during simulation, and the value of this approach is minimal during a simulation because all the attributes are known to the simulation engine. The AC closely examined the paper, and agrees with the reviewers that the approach is very interesting, although perhaps unrealistic for physical robots. However, it does represent significant progress, and the conference should be accepting of bold work. Please improve the paper for the camera ready by incorporating details from the key details in supplemental material into the paper, as outlined by reviewers.